# AGENT PRIORITIZATION WITH INTERPRETABLE RELATION FOR TRAJECTORY PREDICTION

## ABSTRACT

In this paper, we present a novel multi-agent trajectory prediction model, which discovers interpretable relations among agents and prioritize agent's motion. Different from existing approaches, our interpretable design is inspired by the fundamental navigation and motion functions of agent movements, which represent 'where' and 'how' the agents move in the scenes. Specifically, it generates the relation matrix, where each element indicates the motion impact from one to another. In addition, in highly interactive scenarios, one agent may implicitly gain higher priority to move, while the motion of other agents may be impacted by the prioritized agents with higher priority (e.g., a vehicle stopping or reducing its speed due to crossing pedestrians). Based on this intuition, we design a novel motion prioritization module to learn the agent motion priorities based on the inferred relation matrix. Then, a decoder is proposed to sequentially predict and iteratively update the future trajectories of each agent based on their priority orders and the learned relation structures. We first demonstrate the effectiveness of our prediction model on simulated Charged Particles (Kipf et al., 2018) dataset. Next, extensive evaluations are performed on commonly-used datasets for robot navigation, human-robot interactions, and autonomous agents: real-world NBA basketball (Yue et al., 2014) and INTERACTION (Zhan et al., 2019). Finally, we show that the proposed model outperforms other state-of-the-art relation based methods, and is capable to infer interpretable, meaningful relations among agents.

## 1 INTRODUCTION

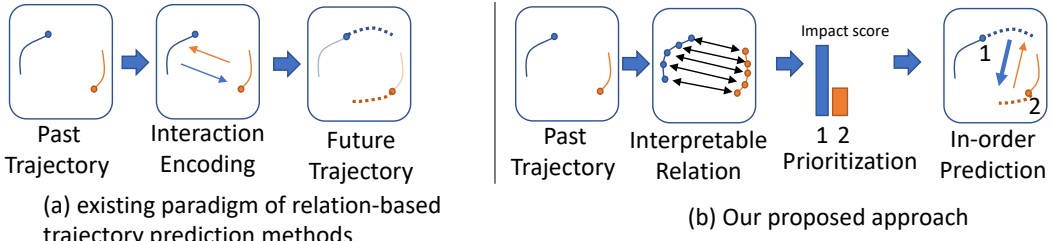

(a) existing paradigm of relation-based trajectory prediction methods

(b) Our proposed approach

Figure 1: Different from the common paradigm on inferring relation for trajectory prediction, our approach aims to learn interpretable relations, prioritize agent motions, and make in-order prediction based on their priorities.

Multi-agent trajectory prediction is an essential component in a wide range of applications from robot navigation to autonomous intelligent systems. While navigating in crowded scenes, autonomous agents (i.e., robots and vehicles) not only themselves interact, but also should have ability to observe others' interactions and anticipate where other agents will move in near future. This ability is crucial for autonomous agents to avoid collisions and plan meaningful machine-human/machine-machine interactions.

Designing a robust and accurate trajectory prediction model has attracted much of recent research efforts. In fact, meaningful reasoning about interactions among agents provides valuable cues to improve the trajectory prediction accuracy, especially in highly interactive scenarios. However, how to

learn/discover meaningful relations among agents from historical motion data to improve prediction accuracy remains a challenging task in recent research. There is a large body of recent research focused on modeling interaction among agents for future trajectory prediction. Some notable research in this field learn interaction features using advanced deep learning techniques such as graph neural networks (Scarselli et al., 2008), social pooling mechanism (Alahi et al., 2016; Gupta et al., 2018), or attention networks (Kamra et al., 2020). These works follow a common paradigm, as shown in Figure 1a, to infer the relations among agents using their historical motions. The main limitation of these approaches is that they lack a mechanism to learn the motion importance of each agent in the scene. In realistically interactive scenarios, it is often that the agent movements implicitly imply that an agent gains higher priority to decide where and when to move, their movements will impact the others in the scene. For example, in driving scenarios, vehicles yield to crossing pedestrians (higher prioritized agents). In the basket ball game, the other players' movements are likely to be conditioned (i.e., impacted ) by the ball-controlling offensive player.

To handle the aforementioned limitations, we propose a new approach, shown in Figure 1b, to prioritize each agent motion based on their interpretable relations. Our model first learns the interaction among agents at each time-step. Inspired by the relation learning method (Fujii et al., 2021) for animal movements, we design a inter-agent encoder that consists of two sub-encoders, each of which represents innate movement and navigation capacities of agents. While the navigation encoder captures the agent relations based on the moving directions (i.e, where to move), the motion encoder infers the relation based on the motion capacity (i.e., how to move). Next, the prioritization module quantifies the importance score (i.e., priority) of each agent by measuring their motion impacts on other agents. Based on the orders of priorities, sequential predictions are made to allow the predicted future trajectories of agents with higher priorities to impact on the lower prioritized ones within their relation structures. In summary, our contributions are:

• We propose a novel prediction pipeline with motion prioritization module to prioritize the importance of each agent based on its motion impacts on other agents within their interpretable relation structures.

• We design an interpretable interaction encoder to capture the agent relations from both navigational and motion perspectives. The relationships among agents are learned interpretably at each observed time step to produce meaningful relation structures for prioritization task.

• We evaluate our prediction model on several highly interactive datasets: Charged Particles, NBA, and INTERACTION. We show that the proposed model is able to learn meaningful interaction features and outperforms state-of-the-art models on these datasets.

## 2 RELATED WORKS

**Multi-Agent Trajectory Prediction**   Multi-agent trajectory prediction is an actively researched problem due to its broad applications in robot planning (Schmerling et al., 2018), traffic prediction (Liao et al., 2018), sport video analysis (Felsen et al., 2017). Recent research have focused on modeling the relations among agents seeking to improve trajectory prediction. In general, existing approaches employ common structures such as graph neural network, social-GAN (Mohamed et al., 2020), attention network, transformer, etc. to learn agent interactions from their motions. Notably, Kosaraju et al. (Kosaraju et al., 2019) on graph attention network (Veličković et al., 2017) to decide how much information to share between agents. Kamra et al. (Kamra et al., 2020) developed a dedicated attention mechanism for trajectory prediction from the inductive bias of motion and intents. Kipf et al. (Kipf et al., 2018) proposed neural relational inference (NRI) proposed by taking the form of a variational auto-encoder;. Jiachen et al. (Li et al., 2020) improved NRI by using graphs that evolve over time. Although these approaches can capture some interactions among agent, they lack mechanism to reason the agent priorities, important feature cues to improve trajectory results.

**Relation Discovery**   Another closely related research theme focuses on discovering the Granger-causal (GC) relationship among agents. SENN (Alvarez Melis & Jaakkola, 2018) is the first self-explanatory network, a class of intrinsically interpretable models, which explains the contributions concepts (i.e., raw inputs) to predictions. SENN was applied to infer GC relationship via generalized vector autoregression model (GVAR) (Marcinkevičs & Vogt, 2021), which captures the GC relationship via coefficient matrices. These models have shown promising performances to learn

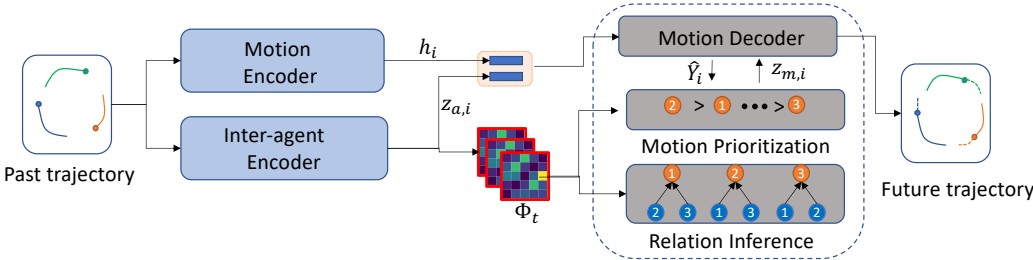

Figure 2: (The overview of our prediction model. Our model consists of three main modules: Motion Encoder (Section 3.1), Inter-agent Encoder (Section 3.3), and Decoder (Section 3.2) with Motion Prioritization and Relation Inference (Section 3.4).

agent interaction in simulated environments, where the GC relationships are known and static. Recently, ABM model (Fujii et al., 2021) was proposed to further extend GVAR model to capture animal interactions such as approaching or repulsing. ABM relies on scientific knowledge and assumes that if an agent goes straight then there is no interactions. For relation discovery in trajectory prediction, (Makansi et al., 2021) analyzed feature attributes to provide insights on the impacts of relation feature and discussed its links to causality inference. Recently, M2I (Sun et al., 2022) learn to predict, but limits to pairs of relations, and only applicable for traffic scenes. Different from these works, we build a new pipeline and extend relation modeling schemes to discover agent relations in real-world interacting scenarios. Based on the learned relation structures, agent motions are prioritized for the purpose of future trajectory prediction.

## 3 METHODOLOGY

In this work, we tackle the problem of multi-agent trajectory prediction. Given the past trajectories $X = [X_1, X_2..., X_N \in \mathbb{R}^{T_o \times d}] \in \mathbb{R}^{N \times T_o \times d}$ of $N$ agents with $d$-dimensional spatial locations in the past $T_o$ time steps, we aim to predict their trajectories $Y = [Y_1, Y_2, ..., Y_N \in \mathbb{R}^{T_p \times d}] \in \mathbb{R}^{N \times T_p \times d}$ in the next $T_p$ future time steps. The overview of our prediction model is shown in Figure 2. Our model consists of three main modules: (1) Motion Encoder: encodes historical motion of each agents. (2) Inter-agent Encoder: encodes interactions among agents; (3) Decoder: infer relations, prioritize motion, sequentially predict future trajectories of agents. Next, we present the details of each module.

### 3.1 MOTION ENCODER

The motion encoder $p(h|X)$ captures the individual movement of agents by learning the hidden motion feature $h \in \mathbb{R}^{N \times d_h}$ conditioned on $X$. We encode the hidden state of each agent $i$ as:

$$e_i = \text{ReLU}(\text{Conv1D}(X_i)) \in \mathbb{R}^{T_o \times C}, \tag{1}$$

$$[o_i^t, h_i^t] = \text{GRU}(e_i), h_i^t \in \mathbb{R}^{d_h}, \tag{2}$$

where $h_i^t$ is the hidden motion state of target agent $i$ at current time step $t$, $o_i^t$ is the output feature of GRU, $C$ is the size of embedded feature $e_i$.

### 3.2 MOTION DECODER

The motion decoder $q(\hat{Y}|h)$ decodes the future trajectories of each agent $i$ from the learned hidden motion feature $h_i^t$. We formulate this decoding process as:

$$[o_i^{t+1}, h_i^{t+1}] = \text{GRU}(h_i^t), \tag{3}$$

$$[\Delta\mu_i^{t+1}, \sigma_i^{t+1}, p_i^{t+1}] = \text{fc}(o_i^{t+1}); \mu_i^{t+1} = \mu_i^t + \Delta\mu_i^{t+1}, \tag{4}$$

$$Y_i^{t+1} \sim \mathcal{N}(\mu_i^{t+1}, \sigma_i^{t+1}, p_i^{t+1}) \in \mathbb{R}^{d_h}, \tag{5}$$

where fc is a fully connected layer. To cope with multi-modal nature of future trajectories, we predict a bivariate Gaussian distribution $\mathcal{N}(\mu_i^t, \sigma_i^{t+1}, p_i^t)$ in each future time step $t \in \{t_0 + 1, ..., t_0 + T_p\}$,

where $\mu_i^{t+1} = (\mu_x, \mu_y)_i^{t+1}$, $\sigma_i^{t+1} = (\sigma_x, \sigma_y)_i^{t+1}$, $p_i^t$ are the mean, standard deviation, and correlation coefficient. We then randomly select $K$ samples from the distribution for the final multiple trajectories prediction. Due to complex interactions among agents, relying on historical motions is not adequate for accurate predictions. Thus, in the next stages, we extend the decoder to incorporate relation features from inter-agent encoder and from the motion prioritization module. The decoder is formulated as $q(\hat{Y}|h, z_a, z_m)$, where $z_a$ and $z_m$ are relation features from inter-agent encoder and motion prioritization, which will be presented next.

## 3.3 INTER-AGENT ENCODER

The encoder $p_a(z_a, \Phi|X)$, depicted in Figure 3, learns interpretable relation features $z_a$ given the past locations of all agents. It also produces relation matrices (i.e., coefficient matrices) $\Phi$, consisting of relation matrix $\Phi^t \in \mathbb{R}^{N \times N}$ at each observed time step $t \in \{t_0 - T_o + 1, ..., t_0\}$. $\Phi_{i,j}^t$ is an asymmetric square matrix, which elements indicate the motion impact of an agent on another. Thus, the value of $\Phi_{i,j}^t$ is high if there is strong motion impact from agent $i$ to $j$. In other words, agent $i$ contains rich information that improves future prediction of agent $j$. Conversely, $\Phi_{i,j}^t = 0$ indicates no relation $i \rightarrow j$. The relation matrices are learned during the training process via inducing sparsity loss, introduced in Section 3.4.

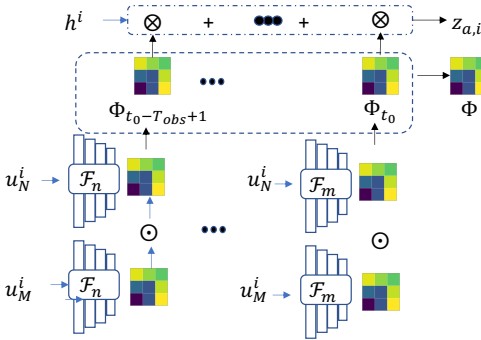

Figure 3: The architecture of Inter-agent encoder.

To learn interpretable and meaningful relation matrix, we design an disentangled inter-agent encoder, which consists of an interaction-based navigation encoder $F_n(\Phi_n|X)$ and an interaction-based movement encoder $F_m(\Phi_m|X)$. Our design is inspired by that autonomous agents innately possess common navigation and movement functions, which enable them an ability to plan the future movements. While the navigation encoder captures the relation based on the future directions of agents, the movement encoder infers the relation based on the strength of their movements. Specifically, we implement our inter-agent encoder following the concepts of self-explanatory neural networks SENN(Alvarez Melis & Jaakkola, 2018), and its extensions to to discover causal relationships (GVAR (Marcinkevičs & Vogt, 2021), ABM (Fujii et al., 2021)). These networks consists of a link function $g(\cdot)$, basis concepts $\psi(x)$, and explainable function $\theta(x)$ to each concept to the predictions. The general form is:

$$f(x) = g(\theta(x)_1 \psi(x)_1, ..., \theta(x)_u \psi(x)_u). \tag{6}$$

In our work, we consider the concepts $\psi(x)$ being motion feature $h_i$, the relation matrix $\Phi_t$ is the result of explainable function $\theta(x)$, and $g(\cdot)$ is the sum. Equation 6 can be written as:

$$z_{a,i}^t = \sum_{t=t_0-T_o+1}^{t_0} \Phi^t h_i^t + \epsilon_t, z_i \in \mathbb{R}^{d_a}. \tag{7}$$

$\Phi^t$ can be decomposed to $\Phi_m^t$ and $\Phi_n^t$; that is $\Phi^t = \Phi_m^t \odot \Phi_n^t$, where $\odot$ is element-wise multiplication, $\epsilon_t$ represent independent noise. Each matrix represents the agent relationships from motion and navigational perspectives. Formally, $\Phi_m^t$ and $\Phi_m^t$ are learned from motion and navigation functions as: $\Phi_m^t = F_m^t(X^t) \in \mathbb{R}^{Nd_i \times (N-1)d_i}$ and $\Phi_n^t = F_n^t(u_n^t) \in \mathbb{R}^{Nd_i \times (N-1)d_i}$, where $u_n^t = [X^t, r^t] \in \mathbb{R}^{d_u}$ is the input feature, concatenating the observed location $X^t$ of all agents and its relative locations to others $r^t = [\Delta X_{i,j(j \neq i)}]$. $F_m$ and $F_n$ are neural networks. Equation 7 makes our relation modeling to capture agent relations in each observed time step, which is different from NRI (Kipf et al., 2018) and dNRI (Graber & Schwing, 2020). We note that ABM (Fujii et al., 2021)) also shares a similar structure, but with limitations that hinder the ability to extend SENN to work in human interactive scenarios. The key differences are highlighted below:

(1) ABM model designs a specific navigation function $F_n$ to capture the relationship of few animal species, where there relationship occurs within a close distance (i.e., local scope). Thus, this model does not generalize well to other real-world applications, where the relation could be occurred in further distance. In this work, we extend the navigation functions to captures relations in both local and global scopes as: $F_n(u^n) = \varsigma_\alpha F_n^l(u_n^l) + (1 - \varsigma_\alpha) F_n^g(u_n^g)$, where $F_n^l$ and $F_n^g$ are local and global navigation functions, which capture the relations within local areas and entire scene, respectively. The contribution of each function is weighted by $\varsigma_\alpha$, a sigmoid function with learnable parameter $\alpha$; $\varsigma_\alpha \in [0, 1]$. We present the implementation and impacts of these functions in the supplementary material.

(2) While previous models (Fujii et al., 2021; Marcinkevičs & Vogt, 2021) infer the causal relationships without focusing on long-term trajectory prediction task. In contrast, our work focus on learning interpretable relation, which is meaningful for motion prioritization in the later stage and the entire pipeline is trained to improve the multi-step future prediction.

### 3.4 MOTION PRIORITIZATION

The motion prioritization module $p_m(z_m, r | \Phi, \hat{Y})$ learns to prioritize the impact of the agents based on the learned relation matrix from Inter-agent Encoder. It also produces the future-trajectories conditioned relational features $z_m$ for each agent. We present the details of each module below:

**Future-Conditioned Relation Inference**  Given the learned relational matrix $\Phi_t$ at each observed time step $t \in \{t_0 - T_o + 1, ..., t_0\}$, we quantify the relation matrix among agents $S \in \mathbb{R}^{N \times N}$ as:

$$S_{i,j} = \frac{1}{T_o} \sum_{t_0 - T_o + 1}^{t_0} \kappa(t, t_0)(||\Phi_{i,j}||_2^2), \tag{8}$$

$$w_{i,j}^r = \frac{S_{i,j}}{\sum_{j=0, j \neq i}^{P} S_{i,j}}; w_{i,j} \in [0, 1]; \sum_{j \neq i}^{N} w_{i,j} = 1, \tag{9}$$

where $\Phi = [\Phi^{t_0 - T_o + 1}, ..., \Phi^{t_0}]$, consisting the relation matrix at each observed time step, and $\Phi^t = \Phi_n^t \odot \Phi_m^t$. Due to the dynamic movements of agents, the recent relation matrix is more relevant to predict future time steps compared to those in further past. To achieve this, we design a temporal kernel $\kappa(t, t_0) = \varsigma_{a_r}(1/|t_0 - t| - d_{th})$ with $\varsigma_{a_r}$ is a sigmoid function with gain $a_r$. The kernel *emphasizes* the relation feature as it closer to the current time step, and weaken those that is away. Next, we construct the future-conditioned relation feature $z_{m,i}$ of target agent $i$ as:

$$\hat{h}_j = \text{GRU}(\text{Conv1D}(\hat{Y}_j)) \in \mathbb{R}^{d_h}, \tag{10}$$

$$z_{m,i} = w_{i,j}^r \times \hat{h}_j \in \mathbb{R}^{d_z}. \tag{11}$$

**Motion Prioritization**  The intuition is that an agent gains higher priority when its motion is less affected by other's motion and more impacts on others. In fact, the impact score can be measured from the relation matrix. We associate each agent with a priority score $m_i = \sum_{j=0, j \neq i}^{j=N} S_{i,j}$. Based on the priority score, we propose to predict future trajectories in order of their priorities. This is to allow the predicted trajectories of higher prioritized agents have impacts on the ones with lower priorities. Considering motion prioritization, we can extend the motion decoder as $q(\hat{Y} | h, z_a, z_m)$. In fact, the predicted trajectories of each agent can be updated (i.e., refined) multiple times to encourage the self-corrected among agent's future trajectories and to fully utilize the future-conditioned relation features. To implement this idea, we iteratively decodes the future trajectories multiple times defined by $N_s \in \mathbb{R}$. Intuitively, if $N_s = 0$, then there is no updates on predictions. In other words, the motion prioritization does not have effects. Otherwise, there will be $N_s$ loop over the prioritized list $\mathcal{O} = \{i, j, ..., N\}$, where $m_i \geq m_j$. The details of decoding procedure and the impacts of $N_s$ on prediction accuracy are presented in our supplementary material.

**Loss Function**   We jointly train the proposed models by using the following losses:

$$\mathcal{L}(\theta) = \sum^{T_p} \mathcal{L}_{pred}(\hat{Y}_t, Y_t) + \lambda \sum^{T_o} \mathcal{L}_{sparsity}(\Phi), \tag{12}$$

$$\mathcal{L}_{pred} = \min_k(||Y_i - Y_i^k||_2), \tag{13}$$

$$\mathcal{L}_{sparsity} = \frac{1}{T_o}(\alpha||\Phi||_1 + (1-\alpha)||\Phi||_F^2), \tag{14}$$

where $\mathcal{L}_{pred}$ is the min-over-k mean-square-error (MSE) prediction loss, which encourages diversities among $K$ predictions sampled from the predicted Gaussian distribution. $\mathcal{L}_{sparsity}(\Phi_t)$ is sparsity-inducing penalty term used to learn the relation matrix. In our implementation, we use the elastic-net-style penalty term (Nicholson et al., 2017) (Equation 14) with $\alpha = 0.5$, and $||\cdot||_F$ is the Frobenius norm.

## 4 EXPERIMENTAL RESULTS

We evaluate our model on simulated Charged Particles (Kipf et al., 2018) dataset and two complex interaction datasets, which commonly favors the applications of robot navigation and autonomous system: NBA (Yue et al., 2014) and INTERACTION (Zhan et al., 2019).

**Charged Particles**   Charged particles (Kipf et al., 2018) is a simulated deterministic system, which is controlled by simple physics rules. In each scene, there are 5 charged particles. Each particle has either a positive or negative charge with equal probability. Particles with the same charge repel each other, and vise versa. We set $T = 100$ and $T_o = 80$ for each scene. We generate 50K scenes for training, and 10K each for validation and test respectively.

**NBA Dataset**   This dataset contains tracking data from the 2012-2013 NBA season, and is provided in (Li et al., 2021). The dataset consist of trajectories of a ball and 10 players from both teams (i.e., 5 players each team). We preprocess the data such that each scene has 50 frames and spans approximately 8 seconds of play, and the first 40 frames are historic, i.e. $T = 50$ and $T_o = 40$.

**INTERACTION**   It consists of different realistic and interactive driving scenarios in roundabout, un-signalized intersection, signalized intersection, merging and lane changing. In total, the dataset is collected from 11 locations using drones or fixed cameras. We follow the same train/validation/test splits as proposed by (Zhan et al., 2019). We set $T_o = 30$ frames (3 seconds), $T_p = 10$ frames (1 second).

The evaluations are made in comparison with state-of-the-art models: Social-GAN (Gupta et al., 2018), Fuzzy Query Attention (FQA) (Kamra et al., 2020), Dynamic Neural Relational Inference (dNRI) (Graber & Schwing, 2020), and GRIN (Li et al., 2021). We evaluate our model using the commonly-used metrics: Average Displacement Error (ADE) and Final Displacement Error (FDE), which are the $L2$ distance between the ground truth and predicted trajectories, and the $L2$ distance between the ground truth final destination and predicted final destination. Following GRIN (Li et al., 2021), we report the best-of-100 displacement error of each trajectory/destination. We present the details of comparison models and implementations in the supplementary material.

### 4.1 QUANTITATIVE RESULTS

We provide quantitative comparisons with related methods on Charged Particles and NBA datasets in Table 1. We follow the same experiment setups used on GRIN (Li et al., 2021). Our approach outperforms relation modeling methods (NRI (Kipf et al., 2018), dNRI (Graber & Schwing, 2020)) in ADE/FDE, which demonstrates the efficacy of discovering agent relations in each observed time step. Additionally, our model also outperforms state-of-the-art model (Kamra et al., 2020) and interpretable model (Li et al., 2021) on both datasets by large margins. It concludes that the proposed prioritization module plays a pivot role in improving prediction accuracy. We perform the ablation study (see Table 2) to investigate the impact of each model component. The brief description of each module is as follows: $\boldsymbol{E}$ using only motion encoder/decoder, $\boldsymbol{F_m}$ is the motion encoder; $\boldsymbol{F_n}$

| Model | Charged Particles ADE/FDE | NBA ADE/FDE |
|---|---|---|
| NRI (Kipf et al., 2018) | 0.63/1.30 | 2.10/4.56 |
| dNRI (Graber & Schwing, 2020) | 0.94/1.93 | 2.02/4.52 |
| FQA (Kamra et al., 2020) | 0.82/1.76 | 2.42/4.81 |
| S-GAN (Gupta et al., 2018) | 0.66/1.25 | 1.88/3.64 |
| GRIN (Li et al., 2021) | 0.52/1.09 | 1.72/3.59 |
| **Ours** | **0.42/0.72** | **1.66/2.89** |

Table 1: Comparisons with Other Relation based Trajectory Prediction Models.

| Components | | | | | Charged Particles ADE/FDE | NBA ADE/FDE |
|---|---|---|---|---|---|---|
| $E$ | $F_m$ | $F_n$ | $Re$ | $Pri$ | | |
| ✓ | | | | | 0.51/0.96 | 1.83/3.46 |
| ✓ | | ✓ | | | 0.50/0.94 | 1.89/3.52 |
| ✓ | ✓ | | | | 0.46/0.88 | 1.87/3.50 |
| ✓ | ✓ | ✓ | | | 0.45/0.87 | 1.87/3.50 |
| ✓ | ✓ | ✓ | ✓ | | 0.48/0.77 | 1.84/2.99 |
| ✓ | ✓ | ✓ | ✓ | ✓ | **0.42/0.72** | **1.66/2.89** |

Table 2: Ablation Study on Charged Particles and NBA Dataset.

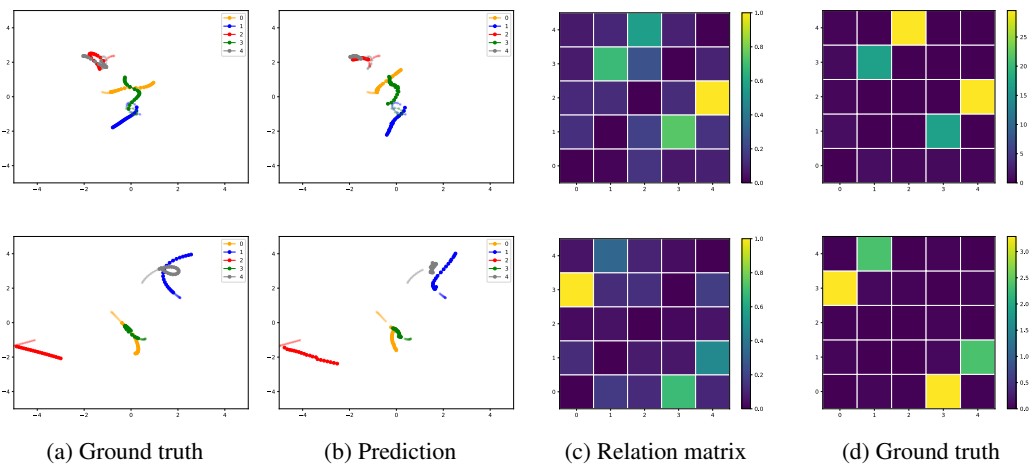

(a) Ground truth     (b) Prediction     (c) Relation matrix     (d) Ground truth

Figure 4: Qualitative results on Charged Particles dataset. Darker colours corresponds to weak relation between agents in relation matrix.

is the navigation encoder, **Re** is future-conditioned relation inference, **Pri** is motion prioritization module. We can see that the relation-based navigation encoder(i.e., $E + F_n$ ) and relation-based motion encoder (i.e., $E + F_m$ ) show their positive impacts when reducing the prediction errors on Charge Particles compared to the variant only individual motion encoder $E$. However, it is interesting to observe that these variants worsen the results on NBA sport dataset. It is reasonable because the NBA consist of mostly realistic and complex interactions among players. The players make strategy movements based on the future movements of other higher prioritized players. This explains why adding relational inference (i.e., module $Re$) to condition one player's movements on other's future trajectories helps reduce the prediction errors. Interestingly, our full model with **Pri** (last row) shows best results overall. This indicates that each player has different motion impact that affects other players' movements in NBA sport games.

## 4.2 QUALITATIVE RESULTS

We show the qualitative results consisting of the predicted trajectory and learned relation matrix. In Figure 4 the charged particles preserve very interesting swirling interactions, such as among two particles 2 and 4 in the first row, and two groups of swirling particles (1,4), and (0,3) in second row. The learned relation matrices show that these interactions can be captured in the corresponding squares (1,3), (3,1), (4, 2), (2,4) for the first scenario, and (3,0), (0,3) (1,4), (4,1) for the second scenario. We note that the stronger relation among agents, the more yellowish (i.e., $S_{ij} \rightarrow 1$) in the corresponding square in the relation matrix, while blue ($S_{ij} \rightarrow 0$) indicates there is no relation.

In NBA dataset, we observe several interesting interaction scenarios, shown in Figure 5. In the first row, there are several notable interaction pairs (2,7), (0, 5), (4,9) captured in our relation matrix. It can be observed that in this scenario player 7 is following along with player 2, moving from one side to another side of the court. At the same time, players 5, 9 play defense against players 0 and 4, respectively. In second row, our model learns there are group of motion among players (0, 6, 8, 1, 7, 3), where player 0 has dominant motion impact on other players. In the last row, there are two

| Models | ADE 1/20/100 | FDE FDE 1/20/100 |
|---|---|---|
| GRIN (Li et al., 2021) | 1.80/1.27/0.67 | 3.31/2.33/1.26 |
| **Ours** | **1.46/1.16/0.61** | **2.69/2.11/1.22** |

Table 3: Quantitative results on INTERACTION datasets with different number of prediction samples 1/20/100.

main groups (1,7) and (2,9, 4 ,5 6,), especially player 9 plays large attention to player 2 as player 9 is defending against player 2, who is trying to score.

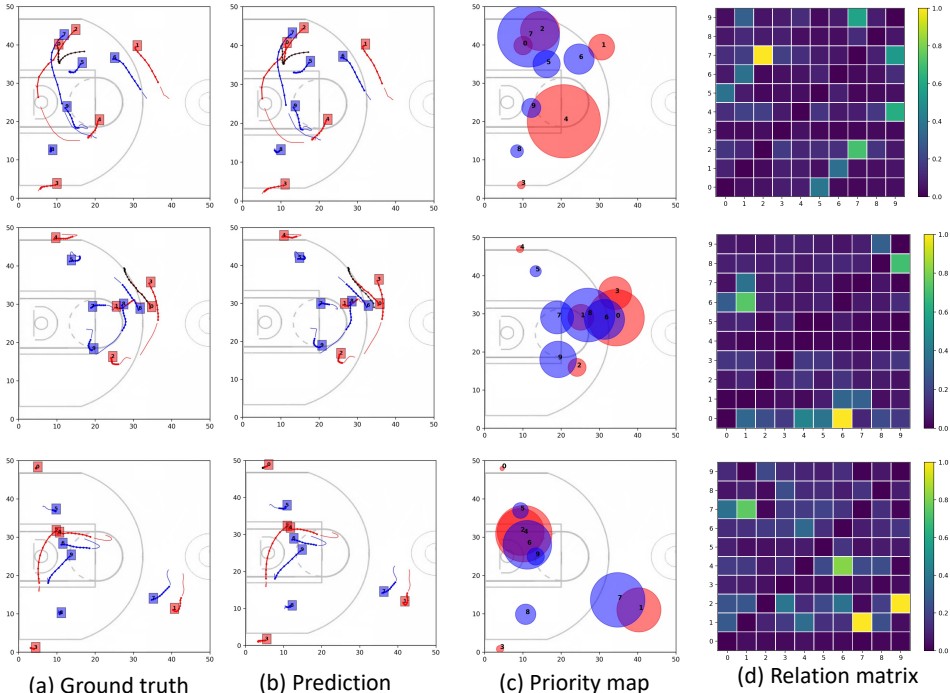

(a) Ground truth    (b) Prediction    (c) Priority map    (d) Relation matrix

Figure 5: Qualitative results on NBA dataset. Offensive team (i.e., controlling the ball) is colored in red, defensive team is colored in blue. The ball is colored in black. Solid line represents observed trajectory. Dotted line represents predicted trajectory. In priority map, the larger circles represent higher priority scores of agents.

## 4.3 RESULTS ON INTERACTION DATASET

In this section, we present our evaluation on INTERACTION dataset in comparisons with GRIN (Li et al., 2021) on different number of prediction samples: 1, 20, 100, as reported in Table 3. Both models are trained with 100 epochs. Overall, our model outperforms GRIN model in both ADE/FDE in different number of predicted trajectories. Finally, we provide visualization on different driving scenarios such as intersection, roundabout and merging in INTERACTION dataset, shown in Figure 6. Generally, interaction types such as following, or yielding are very common in driving scenarios. The results show that our model is able to capture these types of interactions in several scenarios. As we can see that in first row (intersection scenarios), agent 2 is waiting for agent 0 and 1 to pass before it can turn left. Meanwhile, agent 1 makes a slight interaction (i.e., yield) to agent 3. In the roundabout scenarios (second row), the most notable interaction is between 2 and 4. In this scenario, agent 2 is waiting for agent 4 to cross, before it can go into the roundabout section. In the last row, there is interesting merging scenarios from 4 to 3. and agent 3 following 2 when there merge from

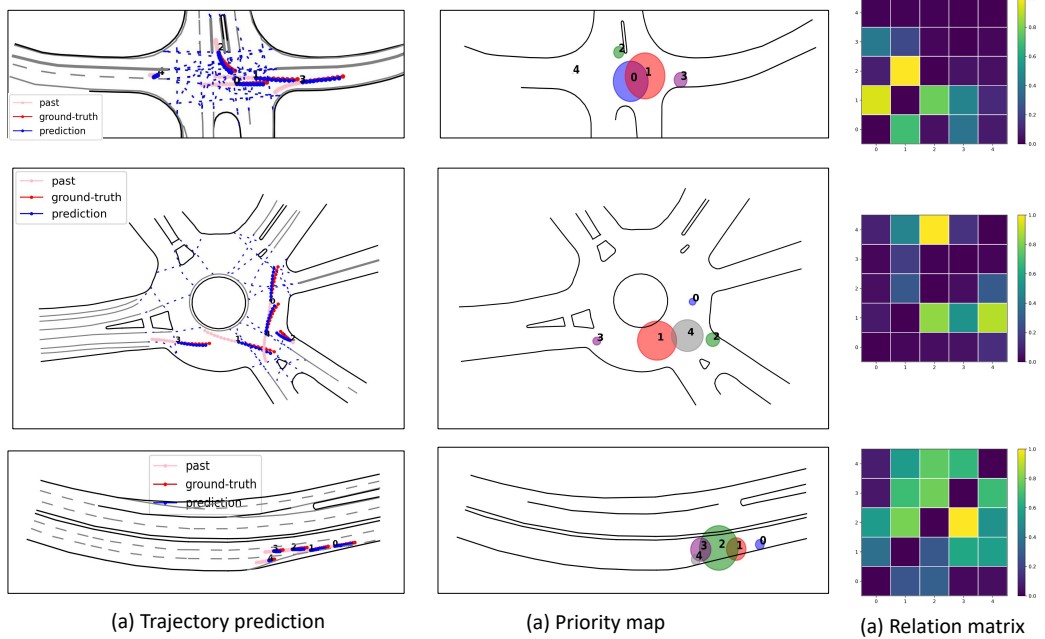

(a) Trajectory prediction       (a) Priority map       (a) Relation matrix

Figure 6: Visualizations from scenarios: intersection (first row), roundabout (second row), and merging (third row) in INTERACTION dataset.

3 to 2 lanes. In addition, the priority map for each scenario indicates the agent's importance, which provides meaning cues for improving trajectory prediction.

## 5 LIMITATIONS

Although we have demonstrated that the proposed model has successfully captured several inter-action behavior in the given datasets, discovering causalities in realistic scenarios such as in au-tonomous driving is still challenging. As a possible solution, exploring scientific knowledge about traffics and the priorities of different agent types could be incorporated to improve the model accu-racy. Secondly, our model discovers relations and prioritization from bird-eye views, which provides a global-view relation agents. However, robots or ego-vehicles also observe the relations with sur-rounding agent from the front camera views. Thus, the future work of considering fusing relations from different views can be explored to strengthen the relation features. Lastly, discovering relation and trajectory prediction (if the identity of subject is known, which is not the case in our work) can raise issues regarding privacy sensitivity.

## 6 CONCLUSION

In this paper, we introduce a novel multi-agent trajectory prediction. Our model discovers inter-pretable interaction feature from historical trajectories of all agents. We also developed motion prioritization module, which prioritize those agents which have higher impacts on other future mo-tions. Based on the prioritized scores, the decoder makes sequential prediction of each agents with iterative predictions. In broader impacts, our prediction could be used to infer relations in other scenarios such as human motions, or inferring robot-human interactions and predictions.

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

# SUPPLEMENTARY MATERIALS:
# AGENT PRIORITIZATION WITH INTERPRETABLE RELATION FOR TRAJECTORY PREDICTION

**Anonymous authors**

In this supplementary material, we first provide the details of Inter-Agent encoder (Section 1). We then discuss algorithm of future trajectory decoding process with motion prioritization (Section 2). We further provide the implementation and training details of our model(Section 3) and additional qualitative results (Section 4) on Charged Particles Graber & Schwing (2020), NBA Yue et al. (2014), and INTERACTION Zhan et al. (2019) datasets.

## 1 DETAILS OF INTER-AGENT ENCODER

We introduced in our main paper that the Inter-Agent encoder consists of movement function $F_m$ and navigation function $F_n$, which capture agent relations from motion and navigation perspectives, inspired by animal movements Nathan et al. (2008); Fujii et al. (2021). Specifically, each function produces a relation matrix ($\Phi_m$ and $\Phi_n$), where each element of $\Phi_m$ (produced by $F_m$) represents the amplitude of interaction, and each element of $\Phi_n$ (produced by $\Phi_n$) represents the sign of interactions (e.g., positive : two agents approach, negative: two agents repulse). To further extend the model to generalize well to real-world robot applications, where the relation could be occurred in further distance. We implement our navigation function to capture interaction from local and global scopes, which capture the relations within local areas and entire scene, respective. Thus, $F_n$ can be decomposed into local function $F_n^l$ and global function $F_n^g$. The local function $F_n^l$ capture interaction with emphasies given a pre-defined thershold $d_{ignore}$ as follows:

$$F_N^L(u_n) = \varsigma_{a_d}\left(\frac{1}{||r^{i,j}||_2} - d_{ignore}\right)(\varsigma_{a_v}(v^{i,j} - \frac{1}{2}) \times 2,\tag{1}$$

while the global function capture the interaction in entire scene (without threshold).

$$F_N^G(u_n) = \varsigma_{a_d}\left(\frac{1}{||r^{i,j}|| * 2}\right)(\varsigma_{a_v}(v^{i,j} - \frac{1}{2}) \times 2,\tag{2}$$

where $\varsigma_{a_d}$ and $\varsigma_{a_v}$ are sigmoid functions with gain $a_d$ and $a_v$, respectively; $d_{ignore}$ is a threshold for defining the local area. Thus, if the distance of two agents are greater than $d_{ignore}$, then there is no interaction. $r^{i,j} = r^j - r^i$ is the relative distance of agent $j$ to agent $i$ with $r^i = (x_i, y_i)^T$. $v^{i,j}$ is the velocity of agent $i$ (i.e., $v^i$) in the direction of $r^{ij}$, thus $\varsigma_{a_v}(v^{i,j} - \frac{1}{2}) \times 2$ sign represents of effects of $j$ on $i$. Then, the navigation function is formulated as:

$$F_N(h^i) = \varsigma_\alpha F_N^L(h^i) + (1 - \varsigma_\alpha)F_N^G(h^i),\tag{3}$$

where $\varsigma_\alpha$ is a sigmoid function with learnable parameter $\varsigma_\alpha \in [0, 1]$.

In addition to our ablation studies presented in (Section 4, Table 2 in main paper), we validate the impact of the proposed navigation functions on trajectory prediction. In this experiment, we train/test the different variants of Inter-Agent encoder on Charged Particles dataset. As shown in Table 1, we can observe that local navigation function $F_n^l$ and $F_n^g$ both improves the prediction accuracy. This indicates the important roles of these functions in our model.

## 2 DETAILS OF DECODING ALGORITHM

We present the details of trajectory decoding process with motion prioritization in Algorithm 1. We note that the predicted trajectories of each agent can be updated (i.e., refined) multiple times to

Table 1: Comparisons with other relation based trajectory prediction models.

| Model | Charged Particles | |
|---|---|---|
| | ADE | FDE |
| GRIN Li et al. (2021) | 0.52 | 1.09 |
| E | 0.54 | 1.01 |
| E + $F_m$ | 0.52 | 0.95 |
| E + $F_m$ + $F_n^l$ | 0.50 | 0.94 |
| E + $F_m$ + $F_n^l$ + $F_n^g$ | 0.48 | 0.93 |

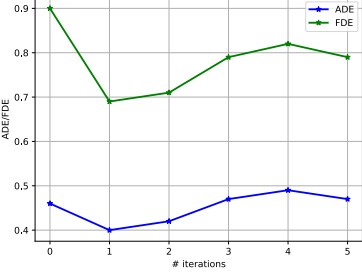 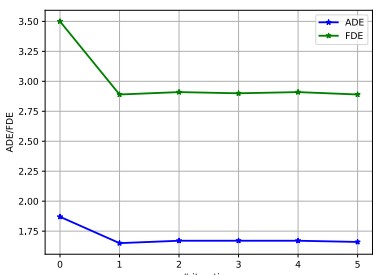

Figure 1: Analysis of number of iterations ($N_s$ on Charge Particles (left) and NBA (right).

encourage the correction among agent's future trajectories and to fully utilize the future-conditioned relation features. To implement this idea, we iteratively decodes the future trajectories multiple times defined by $N_s$. For example, if the prioritized list of agent is $\mathcal{O} = \{3, 2, 1, 4, 5\}$, with 3 is the agent with highest priority, 5 is the lowest one. One iteration (i.e., $N_s = 1$) corresponds to a sequential prediction in order of $3 \rightarrow 2 \rightarrow 1 \rightarrow 4 \rightarrow 5$. This allows the predicted trajectories of agent with higher priority influences on trajectories of agent with lower priority. We study the impact of different number of iterations to update the future trajectories, shown in Figure 1. It is interesting to observe that only one step of refinement is enough to achieve the best prediction on Charged Particles. On NBA dataset, although we achieve best results at 5 iterations, the results also seem to be saturated in the first iteration. This concludes that relational features has been fully shared from the higher-prioritized agent to the lower one in list of prioritized agent in the first iteration.

---

**Algorithm 1:** Algorithm for trajectory prediction with motion prioritization

---

**Input:** List of relational matrices $\Phi = [\Phi^{t_0 - T_o + 1}, \ldots, \Phi^{t_0}]$ in $T_o$ historical time-steps
**Output:** Predicted trajectories $Y = [Y_1, Y_2, ..., Y_N \in \mathbb{R}^{T_p \times d}]$ of all $N$ agents in the next $T_p$ future time steps.

1 Calculate relational weights using Equation 9.

2 Estimate the priority score (i.e., impactness) for each agent. $m_i = \sum_{j=0, j \neq i}^{j=N} S_{i,j}$

3 Rank agents with priority scores in descending order: $\mathcal{O} = \{i, j, ..., N\}$: ordered list of agents, where $m_i \geq m_j$.

4 **for** *step in $N_s$* **do**

5     **for** *i in $\mathcal{O}$* **do**

6         Calculate relational feature $f_i^r$ for target agent $i$ using Equation 10 and 11.

7         Update predicted trajectory

8         $Y_i' = \text{fc}(\text{GRU}([\hat{h}_i, z_{m,i}])$

9 .

---

## 3 IMPLEMENTATION DETAILS

We implement our model using PyTorch Pytorch (2018), which has BSD-style license. We train the model with the Adam Zhang (2018) optimizer with the learning rate of 0.0001 and 100 epochs. The

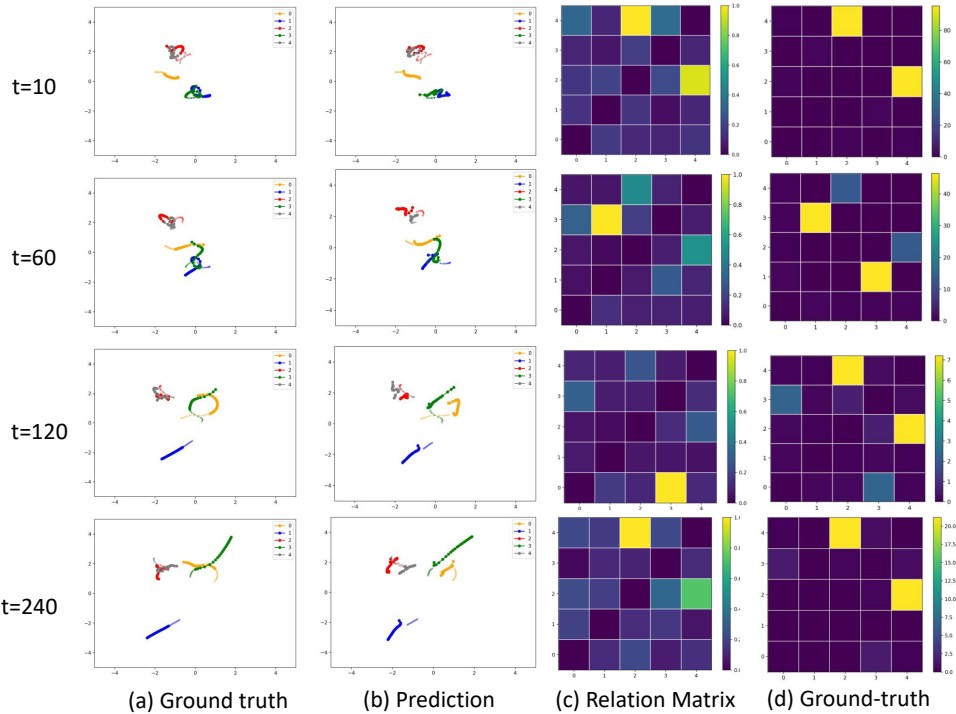

Figure 2: Additional qualitative results on Charge Particles dataset

batch size is 128. For all datasets, we implement the movement encoders with 2 MLP layers, each has hidden size of 50. The hidden size of GRU in both encoders and decoders is 48. In the training loss, we set $\alpha = 0.5$, and $\lambda = 1$.

## 4   ADDITIONAL QUALITATIVE RESULTS

In this section, we present additional qualitative results on three datasets: Charged Particles, NBA, INTERACTION.

**Results on Charged Particles**   Figure 2 shows the trajectory prediction and the learned relation matrix at different time steps. We observe that at different time steps, particles involves in different interactions. For example, at t=10 and t=60, there are mainly swirling interactions between particles (2, 4) and (1, 3). At t=120, particle 3 reaches toward particle 0, and then goes further away at t=240. The learned relation matrix show that these subtle movements and interactions can be captured.

**Results on NBA**   Next, we show predictions and priority scores of players in basketball games (Figure 3). NBA dataset consists of highly complex interactions among players. In the first scenario, player 0 and 4 in offensive team (red) have higher priority score (i.e., higher impact) as the player 0 is passing the ball to player 4; thus, their movements highly influence others' motions. In the second scenario, player 0 holding the ball but does not have large impact; this is because the motions of the ball and player 0 is not changing much. Other players may suspect that the player 0 could pass the ball to other offensive players. Thus, the movements of other players in the red team are more important to observe. Interestingly, player 2 has the most critical move and thus it generates the largest impact on other players.

**Results on INTERACTION**   We further visualize the trajectory prediction, priority map, and relation matrix for two interactive scenarios: merging and roundabout in Figure 4. We can see that our model can capture intriguing relation matrix and learns the motion importance of agent effectively. In the merging scenarios (top row), agent 2 gains highest importance as it is going to merge and have high relations with others agents in front (agent 1), and behind (agent 3). In the roundabout scenario (bottom row), we observe two groups: (1,3) and (0,2,4) on different sides of the roundabout. Interestingly, we can see that the relation between agent 0 (pedestrian) and agent 2 (vehicle) can be

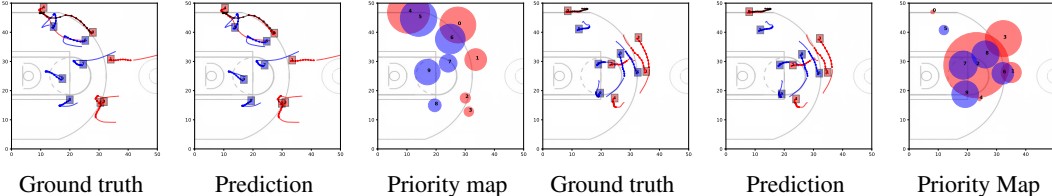

Figure 3: Qualitative results on NBA dataset. Offensive team (i.e., controlling the ball) is colored in red, defensive team is colored in blue. The ball is colored in black. Solid line represents observed trajectory. Dotted line represents predicted trajectory. In priority map, the larger circles represent higher priority scores of agents.

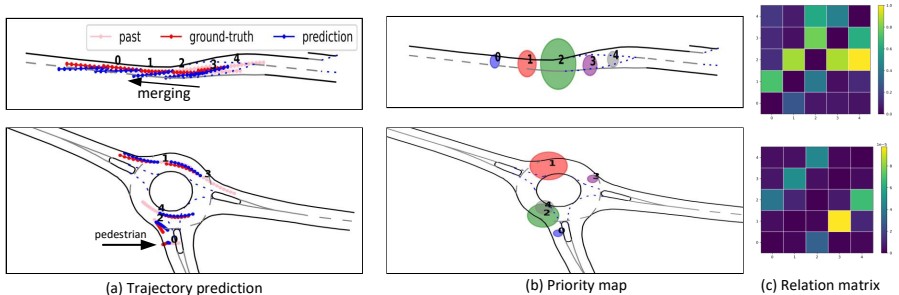

Figure 4: Additional qualitative results on INTERACTION dataset.

captured even though they still far-way. The agent 2 gains most motion priority within the group (0,2,4) because it has relationship with both agent 4 and 0.

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
