# OpenReview forum: "Agent Prioritization with Interpretable Relation for Trajectory Prediction"
_ICLR.cc/2023/Conference — Submitted to ICLR 2023_

### Official Review · Reviewer_8XRE · 2022-10-14

**Confidence:** 3
**Correctness:** 3
**Technical Novelty And Significance:** 3
**Empirical Novelty And Significance:** 2
**Recommendation:** 5

**Clarity, Quality, Novelty And Reproducibility:**

- **Clarity**: Overall, I think that the paper is clear, although notations could be improved to make it more understandable.
- **Quality**: The quality of the work is ok but can be improved with some more experiments to fully show the significance of the proposed approach.
- **Novelty**: Introducing the concept of priority for trajectory forecasting seems to be a novel idea as far as I checked.
- **Reproducibility**: Currently not very reproducible. Although some implementation details are provided in the paper, a complete codebase would be necessary to fully reproduce the published results. The proposed method involves a generative process (Eq 5) and its performance may vary depend on random seeds.

**Strength And Weaknesses:**


## Strong points
- Overall the paper is clear and nicely written.
- The proposed work is well motivated. I understood the importance of taking into account the priority in multi-agent trajectory forecasting.
- Overall, the proposed method is technically sound and designed reasonably. Although the proposed method is built on top of some known techniques (standard encoder-decoder-style forecasting + an extension of causal relationship modeling such as SENN, GVAR, and ABM), introducing the concept of priority for trajectory forecasting seems to be a novel idea as far as I checked.
- Extensive experimental evaluation is presented. The proposed method is evaluated on multiple datasets. Its effectiveness is demonstrated compared to multiple baseline methods, and with multiple metrics. The ablation study in Table 2 shows the contribution of each technical component of the proposed method. Qualitative results nicely explain how the proposed method works.

## Weak points
- The paper could have been better if it were clear how large a number of agents the proposed method can deal with. As presented in Section 3.3, the proposed method involves a relation matrix with the size of NxN that becomes inevitably huge as the number of agents increases. Moreover, due to the nature of priority-driven forecasting, the decoding part of the proposed method cannot be parallelized. It remains unclear to me how increasing the number of agents affects the forecasting performance and computational cost (or runtime) of the proposed method. As far as I checked, the numbers of agents considered in the experiments were just 5 (Charged Particle and INTERACTION) and 10 (NBA), although there are other datasets with a lot more agents (e.g., ETH/UCY, Stanford Drone Dataset).
- As a relatively minor point, it's not perfectly clear from the current manuscript how is the proposed method better compared to prior work in terms of "interpretability".
- Notations could be improved. It was difficult to understand how \Phi, \Phi_t, \Phi_{i, j}, and \Phi^t_m are related in Section 3.3 and 3.4.


## Additional feedback
- A few typos were found:
  - a inter-agent -> an inter-agent (p.2)
  - we now can -> we can now (p.5)
- It would be better if qualitative results of baseline methods are also provided.

**Summary Of The Paper:**

This paper presents a new method for multi-agent trajectory forecasting. Unlike existing methods that just encode inter-agent interactions, the proposed method additionally learns to predict the priority of agents that explains the dependency between agents regarding their future trajectories (i.e., future trajectories are generated in order based on this priority.) The proposed method is evaluated on multiple datasets (Charged Particles, NBA Dataset, and INTERACTION) and confirmed to outperform multiple state-of-the-art methods in terms of ADE and FDE.

**Summary Of The Review:**

While I acknowledge the novelty of the proposed method, its effectiveness remains questionable to me. Specifically, I would like to see the following points clarified.

1) How is the proposed method scalable for a larger number of agents, in terms of forecasting performance (ADE/FDE) and required computational cost (runtime)?
2) How is the proposed method compared to existing methods in terms of interpretability? I believe that it is another important aspect as "interpretable" is emphasized in the paper. What can be interpreted only by the proposed method and not by any other method?

---

### Official Review · Reviewer_fWdF · 2022-10-24

**Confidence:** 4
**Correctness:** 3
**Technical Novelty And Significance:** 2
**Empirical Novelty And Significance:** 3
**Recommendation:** 5

**Clarity, Quality, Novelty And Reproducibility:**

- Clarity:
High.
This paper is well motivated and well written as well. In general, explicitly modelling interaction between agents is promising and provides interpretable understanding of designed model.

- Quality:
Moderate.
1. Visualization: I found it hard to interpret visualized prediction in Fig.5 in main and Fig.3 in supplementary. Why are there many prediction not associated with any agent? For instance, what is going on with object 6, object 9 and object 7 in prediction column of Fig. 5 (main paper) from top to bottom? How should we interpret these predictions?
2. Results on large dataset: Many existing work validate their ideas on large outdoor dataset. Can the authors explain why not working on large datasets, e.g. NuScenes or Waymo or ArgoVerse, but choose INTERACTION?
3. Diverse predictions: Do the authors think about the diversity in their prediction? And will you consider this in future work?
4. Sparsity: It does not seem to me that the sparsity loss truly enforces the sparsity. Rather, it learns the weight. I believe this is also supported by visualizations, e.g. the learned relation matrix in Fig.4 and Fig.5.

- Novelty:
Moderate to low.
The explicit modelling process is formulated as a relation matrix learning problem. Though the authors clarified the differences between the proposed method and literature, e.g. SENN(Alvarez Melis & Jaakkola, 2018), GVAR (Marcinkeviˇcs & Vogt, 2021), ABM (Fujii et al., 2021), I found the differences are not significant. Or in another word, not significant enough to support a ICLR paper.

- Reproducibility:
Low.
The architecture details, e.g. number of layers or layer design, are missing, which also makes it hard to judge whether comparison to existing work is fair or not.

**Strength And Weaknesses:**

 Strength:
1. This paper is well-written and easy to follow
2. Novelties are well-explained and motivations are clear
3. The proposed method achieves SOTA performance

Weakness:
1. In terms of novelty, the relationship modeling is minor from my perspective.
2. The visualization, or qualitative results, seems to be problematic.
3. Architecture details are missing, which makes it hard to re-produce the reported results.
4. Extensive results on larger datasets, e.g. NuScenes or Waymo, are expected.
(See below for more detailed explanations for weakness)

**Summary Of The Paper:**

This paper proposes a new method for multi-object trajectory prediction. Specifically, the proposed method consists of two different encoders and one motion decoder. The conventional motion encoder  inputs past trajectories and outputs the features and hidden motion state of each agent. And the motion decoder receives hidden states and decodes the future trajectories of each agent. Compared to existing methods, the authors introduce a relation matrix that explicitly measures the impact between each agent, as additional inter-agent encoder. To be specific, the inter-agent encoder consists of a navigation and a movement encoder and both of them are implemented in self-explanatory manner, as appeared in literature.

The authors validate their ideas on three dataset, one simulated dataset and two real ones. Comparing to existing work, such as Social-GAN and GRIN, they demonstrate better performance in terms of ADE and FDE.

**Summary Of The Review:**

Overall, I appreciated the writing, clearness and quantitative results of this paper. However, I have concerns in terms of the novelty and the qualitative results. I am willing to raise my rating if my concerns can be addressed. Thanks.

---

### Official Review · Reviewer_H2is · 2022-10-27

**Confidence:** 5
**Correctness:** 3
**Technical Novelty And Significance:** 3
**Empirical Novelty And Significance:** 3
**Recommendation:** 3

**Clarity, Quality, Novelty And Reproducibility:**

The paper is clear with reasonable quality (some figures like figure 4 needs a bigger font).
The work is novel.

**Strength And Weaknesses:**

Introduction:
- Introduction is clear and work is well motivated based on the example of yielding to pedestrian.

Literature review:
- Literature review is limited in terms of motion prediction methods. The review jumps directly to GNNs methods with focus on attention ignoring other related works that uses GANs, non-attention GNNs.. etc.
- For relation discovery the work of “You mostly walk alone: Analyzing feature attribution in trajectory prediction” is highly related and a proper contribution comparison is necessary.

Methodology:
- The problem definition needs a re-visit, for the example the dimensions of Y is missing.

- 3.1: Reason to use GRU?
- 3.3 The $\Phi$ is an NxN matrix the defines the relationship between agents. Aka it’s an adjacency matrix that defines the importance of agents where agents are graph nodes and $\Phi$ is what defines the edges. Then, what’s the difference between this and ordinary graph attention or a a hand crafter graph edges?
- Page 5: Did you notice that $F_n(u^n)= $ acts as a complimentary filter?
- Motion prioritization: It is still not clear to me how it works? Is there a feedback from higher priority predict agents into the model to predict the lower priority ones? What happens if if there is an out of priority order prediction, reverse priority?

Results:
- Proper introduction of best-of-N ADE/FDE is not present.
- Expected to see results on classic motion prediction datasets such as ETH/UCY, SDD ..
- What about memory footprint and inference speed of the method
- I expected an analysis that shows how higher priority agents results are better than lower priority ones, proving the intent of the prioritization component. Something like this agent(s) ADE/FDE without prioritization is X/Y and this agent(s) is A/B, when the prioritization component is used we see that the first agent(s) ADE/FDE increases while the second agent(s) ADE/FDE decreased.


**Summary Of The Paper:**

The work presents a model for motion prediction. The core component of the method is the prioritization of the agents. Agent with higher priority will have better prediction. The work comes supported with experiments and good analysis.

**Summary Of The Review:**

The paper is interesting, well-motivated.
Extra results and deeper analysis is needed to justify some components and extra literature review is needed.

---

### Decision · Program_Chairs · 2023-01-20

**Decision:**

Reject

**Justification For Why Not Higher Score:**

Reviewers raised a number of serious concerns with the paper, many of which were left unaddressed.

**Justification For Why Not Lower Score:**

N/A

**Metareview: Summary, Strengths And Weaknesses:**


 This paper tackles a multi-agent trajectory prediction problem, utilizing an encoder-decoder mechanism where the encoder summarizes past trajectories via features and decoder predicts future trajectories. The main approach proposed is to further utilize an encoder that represents inter-agent relationships in order to improve such forecasts. Results are shown on three datasets, both simulated and real.

  All of the authors found the paper easy to read and well-motivated, but found significant deficiencies in terms of clarity of results/visualizations, more standard comparisons (outdoor datasets, ETH/UCY, SDD, etc.), computational cost analysis, and especially reproducibility with detailed architectural details missing. The authors only rebutted one review, and after discussion all reviewers agreed that the paper is not ready for publication in its current form.